# DuoLLM: A Dual-Stream Decoupled Visual Language Model for 3D Spatial Reasoning

## ABSTRACT

Recently,Large Vision-Language Models have achieved strong performance in visual recognition tasks, yet their capacity for 3D spatial reasoning remains limited. Such reasoning is critical for downstream applications, including scene understanding and robotic manipulation, where models must infer not only what objects are but also how they are arranged in space. Specificly,two factors contribute to this limitation: (i) a semantic bias, where object meaning tends to dominate over geometric structure, and (ii) the loss of depth cues when 3D environments are projected into 2D images. Consequently, current LVLMs often fail to generalize to tasks that require reasoning about inter-object relations in three dimensions.To address this,this paper propose DuoLLM, a multi-modal architecture that explicitly decouples semantic and spatial processing. The model adopts a two-stream design, where a dedicated spatial stream is equipped with a 3D perception engine. This engine combines (i) a depth estimation module that introduces 2.5D geometric priors for recovering lost geometric information, (ii) a relational attention mechanism that captures object-to-object spatial dependencies for overcoming semantic bias, and (iii) an asymmetric cross-attention module that fuses semantic and spatial features efficiently.On the challenging SpatialScore-Hard benchmark, DuoLLM delivers substantial improvements over widely used open-source LVLMs and approaches the performance of specialized SpatialAgent frameworks. These gains are consistently observed across diverse categories of 3D reasoning, highlighting the effectiveness of the proposed two-stream design and 3D perception engine. Together, these results suggest that explicitly modeling geometric priors provides a promising path to extend LVLMs beyond 2D recognition, advancing toward embodied AI systems capable of spatial reasoning in real-world environments.

## 1 INTRODUCTION

Recent large vision-language models (LVLMs) have made major progress in multimodal understanding(Li et al., 2023; Radford et al., 2021; Jia et al., 2021; Liu et al., 2023), especially in tasks like image captioning and visual question answering(Goyal et al., 2017; Chen et al., 2015). These models show strong potential across many fields, from medical diagnosis(Wang et al., 2022) to everyday human-computer interaction. However, while these models excel at identifying objects in images ("what"), they struggle badly with understanding spatial relationships ("where" and relative positions). When asked "From the viewer's perspective, is object A to the left or right of object B?", most models performance was not satisfactory , despite correctly identifying both objects.

Previous attempts to fix these problems have been inadequate(Hazard et al., 2019; Chen et al., 2024; Cheng et al., 2024). Rule-based spatial templates or synthetic data augmentation only achieve small improvements, while methods using additional 2D spatial encodings fail to capture the inherently 3D nature of spatial relationships. We attribute this to a vicious cycle created by both architectural flaws and input limitations.

The common single-stream architecture leads to systematic "Semantic Prioritization"(Lu et al., 2019; Tsimpoukelli et al., 2021; Yarom et al., 2023). In models, the end-to-end language modeling objective naturally favors global semantic features that help generate text. Under the

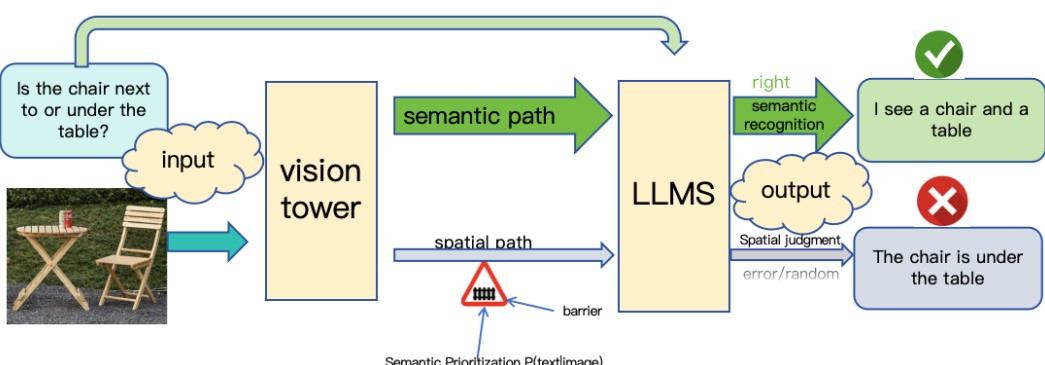

Figure 1: illustrates "Semantic Prioritization" in models

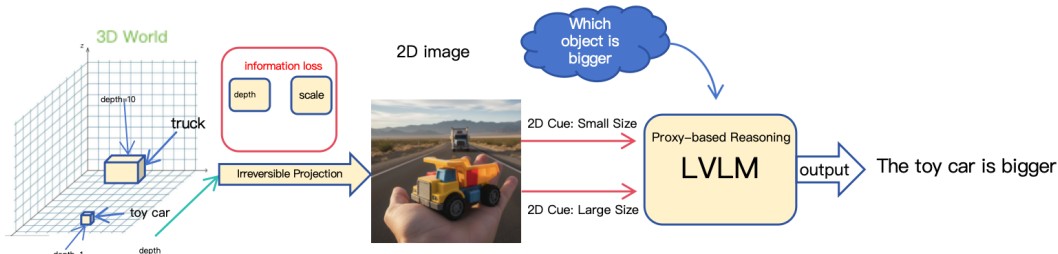

Figure 2: ) highlights 3D Information Loss in models

language modeling objective P(text—image), the model learns to maximize text generation likelihood, naturally favoring high-level semantic features that directly help predict words. Geometric features—essential for spatial reasoning but less useful for immediate text generation—gradually get pushed aside during training(Hazard et al., 2019; Suhr et al., 2019). For example, As shown in Figure1,when asked "Is the chair in the picture under or next to the table?", these models accurately identify both "chair" and "table" but perform near randomly when judging their precise relationship. This happens because the optimization objective doesn't force the model to understand spatial layout, causing geometric features to be systematically marginalized during learning. It's like Similar to "object inventory", rather than a structured scene graph.

This challenge is made worse by the inherent "3D Information Loss" in the input. When visual inputs to LVLMs are irreversible projections from 3D to 2D(Ranftl et al., 2020; 2021), losing crucial information like true scale, depth, and occlusion relationships. This forces models to rely on "proxy-based reasoning" using fragile 2D cues(Luo et al., 2024) (relative size, overlap, perspective lines) rather than understanding true 3D geometry. When these proxies fail—which happens often in real scenes—the model's spatial reasoning breaks down catastrophically. As shown in Figure2 ,A typical failure case shows that when asking models which object is bigger, they make wrong judgments simply because a distant object appears larger in the 2D image. This reliance on ambiguous 2D cues, rather than understanding 3D physical laws, is the fundamental reason for poor reasoning robustness(Yarom et al., 2023).

To break this vicious cycle limiting spatial cognition in LVLMs, we propose a new paradigm that intervenes at both architectural and information source levels: DuoLLM. Our core design principle is: it must use explicit decoupling to create a separate and protected computational space for learning and reasoning about geometric information.

To achieve this, DuoLLM's architecture includes two synergistic innovations. First, to solve the "Semantic Prioritization" problem, we designed a decoupled dual-stream processing framework. This framework extracts deep features for semantic understanding and mid-level features for spatial analysis from middle layer of the visual encoder. Through forced information splitting, we create a dedicated path for geometric information that won't be overwhelmed by strong semantic

signals. Second, to compensate for "3D Information Loss", we equipped the spatial processing stream with an integrated 2.5D perception and reasoning engine. This engine uses a seamless three-stage process to convert external geometric priors into structured knowledge the model can use: it first uses a pre-trained monocular depth estimation network to perceive geometry, reintroducing the third dimension as a 2.5D representation. Next, it explicitly encodes and reasons about relative 3D positions between patches using a novel 3D-aware relative attention bias mechanism. Finally, it employs asymmetric cross-attention to let enhanced spatial features actively fuse with global semantic information, achieving the final binding of structure and concept.

This paper makes three main contributions.

1. we identify and systematically explain the "vicious cycle" in existing LVLMs caused by both "input limitations" and "architectural flaws", providing a new analytical perspective for this research area.

2. based on this analysis, we propose and implement DuoLLM, a novel LVLM paradigm with dual-stream architecture and integrated 3D-aware engine, offering a concrete and viable technical path for models to effectively integrate and use geometric priors.

3. through comprehensive ablation studies, we not only validate the independent value of each proposed component but, more importantly, prove that architectural decoupling and geometric compensation are both indispensable for achieving robust spatial reasoning ability, demonstrating strong synergy between them.

## 2 RELATED WORK

### 2.1 LARGE VISION-LANGUAGE MODELS

The development of Large Vision-Language Models (LVLMs), marked by pioneering work like LLaVA (Liu et al., 2023), MiniGPT-4(Zhu et al., 2023), and InstructBLIP(Dai et al., 2023), has greatly pushed the boundaries of general artificial intelligence. The core design of these models is to align a pre-trained vision encoder with a large language model (LLM) through lightweight projection modules. Follow-up research like LLaVA-1.5 (Liu et al., 2024) and InternVL(Zhang et al., 2023) achieved excellent performance on general dialogue and reasoning tasks by optimizing architecture components and training strategies. However, these improvements mainly focus on data scale, resolution, and connector design, without addressing the fundamental bottleneck of spatial reasoning.

The success of existing LVLMs is mainly limited to the semantic level. Their commonly used architecture paradigm—flattening the 2D grid features from vision encoders into 1D sequences before feeding them to LLMs—is inherited from pure text processing but goes against the natural properties of visual information. This flattening operation destroys the metrically accurate 2D topological relationships in images(Dosovitskiy et al., 2021; Yarom et al., 2023), making it hard for models to rebuild coherent and faithful spatial representations beyond semantic alignment. Therefore, LVLMs have an inherent architectural bias that favors high-level semantic abstraction while sacrificing spatial structural fidelity, directly leading to their limited performance on tasks requiring precise spatial perception.

### 2.2 SPATIAL REASONING IN MULTIMODAL MODELS

How to improve spatial reasoning in LVLMs has become a key research focus. Early approaches relied heavily on external, specialized modules. For example, some work (Zheng et al., 2025)used a "detect-then-reason" pipeline: first locating objects with detectors, then feeding text with bounding box coordinates into LLMs. These methods are limited by detector accuracy and predefined vocabularies, making it hard to handle open vocabulary and abstract concepts. Scene graph-based methods(Zhang et al., 2019; Bourouihiya et al., 2019) also face issues with complex parsing and difficulty in joint optimization with LLMs.

Recent research has shifted toward implicit spatial enhancement within models. For instance, LLaMA-Adapter(Zhang et al., 2024) and LISA(Lai et al., 2024) use adapters or segmentation

decoders to help models handle instructions related to image regions. While effective for 2D grounding and referring tasks, these methods mainly focus on "what is where." In contrast, newer work has started exploring true 3D spatial reasoning, such as SpatialLLM and several 3D-informed models (Chen et al., 2024; Cheng et al., 2024), which try to improve 3D understanding through 3D data, depth estimation, or structured biases. These approaches show that moving from 2D "referring" to 3D "relational reasoning" is becoming a new direction for LVLM development, though current implementations vary widely without a unified solution. Our DuoLLM aims to directly answer the deeper question of "the relative position, orientation, and occlusion relationships between objects A and B in 3D space."

### 2.3 3D PERCEPTION AND STRUCTURED REPRESENTATION

In the broader computer vision field, using depth information to enhance 3D understanding has a long history. While active sensors like LiDAR [53] provide precise depth, monocular depth estimation [44, 45] from single images is more practical for general LVLM applications. Some work has tried integrating monocular depth or 3D-informed data into multimodal models(Lee et al., 2025; Hong et al., 2023), showing the complementary nature of language and geometric information. However, these methods often achieve fusion through feature concatenation or additional token injection, making it hard to truly affect the model's core reasoning process.

Positional encoding(Vaswani et al., 2023) in Transformer architectures provides inspiration for modeling structured information, but mainstream methods are almost entirely limited to 2D or used only within single modalities. Our work makes a key extension here: we propose a 3D relational attention mechanism that transforms geometric relationships like (x, y, depth) into learnable, dynamic attention biases, directly injecting them into cross-modal self-attention computation. This is the first attempt in LVLMs to deeply integrate externally predicted geometric priors into the reasoning core in a structured way, distinguishing it from previous methods that only perform shallow feature fusion.

## 3 PRELIMINARIES

### 3.1 VISION-LANGUAGE MODELS AND THEIR SPATIAL LIMITATIONS

Current Vision-Language Models (VLMs)(Radford et al., 2021; Li et al., 2023; Alayrac et al., 2022) work in a fairly straightforward way: they take visual features from images, align them with text embeddings, and feed both into a language model. Given an input image , models typically use a vision encoder (often CLIP) to extract patch-level features:

$$v = \varepsilon_v(I) \in \mathbb{R}^{N \times D}$$

where N is the number of patches and D is the feature dimension. Here's where things go wrong. To make these features compatible with language models, we have to flatten the 2D feature grid into a 1D sequence. This is problematic—when you reshape $\sqrt{N} \times \sqrt{N}$ patches into a linear sequence, you lose the natural 2D structure. Patches that were neighbors in the image might end up far apart in the sequence(Dosovitskiy et al., 2021; Shaw et al., 2018).

### 3.2 THE DEPTH ESTIMATION COMPONENT

Monocular depth estimation(Ranftl et al., 2021) aims to predict depth maps from single RGB images. Given an image , a depth network produces $D = F_{\text{depth}}(I)$ where each pixel value represents distance from the camera:

$$D = F_{\text{depth}}(I)$$

Recent methods like DPT use Vision Transformers, which handle long-range dependencies better than CNNs. This helps in complex scenes where understanding global context matters for depth prediction.

But here's the challenge: integrating depth into VLMs isn't straightforward. Depth maps are noisy, especially at object boundaries where we need them most. They also lack absolute scale—monocular depth only tells you relative distances, not actual measurements.

### 3.2.1 ATTENTION MECHANISMS IN VISION TRANSFORMERS

Self-attention computes relationships(Vaswani et al., 2023) between all token pairs:

$$\text{Attention}(Q, K, V) = \text{softmax}\left(\frac{QK^T}{\sqrt{d_k}}\right) V$$

Vision Transformers typically add 2D positional encodings so the model knows where each patch comes from in the image. But these encodings only capture coordinates—they don't encode depth or 3D relationships.

For genuine spatial reasoning, we need attention mechanisms that understand 3D structure. Some work has explored relative position encodings, but mostly within single modalities. When it comes to cross-modal attention between vision and language, geometric relationships are usually ignored entirely. The model might know that two patches are "related" but not whether one is in front of, behind, or beside the other.

### 3.2.2 MULTI-STREAM PROCESSING

The idea of separate processing streams comes from neuroscience—the brain has distinct "what" and "where" pathways(Goodale & Milner, 1992). In computer vision, we can implement this by splitting visual processing(Simonyan & Zisserman, 2014):

1. Semantic stream: Focuses on object identity and scene understanding ("what")
2. Spatial stream: Handles geometric layout and relationships ("where")

In practice, this means extracting features from different layers of the vision encoder. Early layers retain spatial detail but lack semantic understanding. Later layers are semantically rich but spatially compressed—the model knows there's a "cat" but might have lost track of exactly where its paws are.

The hard part is combining these streams. Simple concatenation doesn't work well because semantic and spatial features have different properties. Cross-attention seems promising, but current implementations don't really handle the geometric aspects properly. You end up with a model that knows both "cat" and "position (x,y)" but still can't reason about whether the cat is on the table or under it.

## 4 METHODS

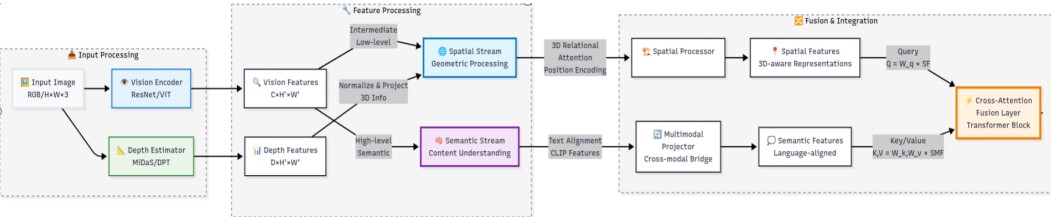

Figure 3: shows overall framework.

### 4.1 DUAL-STREAM PROCESSING

Most vision-language models use the final layer of a vision encoder because it has the best semantic features. But these features have lost most spatial information through pooling and attention

operations. Using earlier layers would keep spatial info but wouldn't understand objects well enough.

So we use both. We take a pre-trained CLIP ViT-L/14 encoder and extract features from two different layers:
**Semantic features** from the layer -2:

$$v_{\text{sem}} = \varepsilon_v(I)^{[-2]} \in \mathbb{R}^{N_s \times D_s}$$

**Spatial features** from layer -16:

$$v_{\text{spat}} = \varepsilon_v(I)^{[-16]} \in \mathbb{R}^{N_p \times D_p}$$

We picked layer because it seemed to work best in our experiments - it still knows about objects but hasn't thrown away all the spatial information yet.

### 4.2 ADDING 3D UNDERSTANDING

The spatial stream needs extra processing to understand 3D structure. We do this in three steps: add depth, add 3D-aware attention, then combine with semantics.

#### 4.2.1 ADDING DEPTH

First, we need depth information. We use DPT (a pre-trained depth estimator) to get a depth map from the RGB image:

$$M_d = DPT(I) \in \mathbb{R}^{H \times W}$$

This depth map is the wrong size for our features, so we resize it:

$$M_d' = resize(M_d) \in \mathbb{R}^{\sqrt{N_p} \times \sqrt{N_p}}$$

Different images have different depth ranges, so we normalize to [0,1]:

$$M_d'' = \frac{M_d' - \min(M_d')}{max(M_d') - \min(M_d')}$$

Then we project these depth values into feature space using a 2-layer MLP:

$$f_d = MLP(flatten(M_d'')) \in \mathbb{R}^{N_p \times D_p}$$

Now we need to combine depth with visual features. We tried adding them but concatenation worked better:

$$v_{concat} = [v_{spat} \| f_d] \in \mathbb{R}^{N_p \times 2D_p}$$

We use another MLP to fuse them:

$$v_{spat}' = FusionMLP(v_{concat}) \in \mathbb{R}^{N_p \times D_p}$$

Plus a layer norm for stability:

$$v_{spat}'' = LayerNorm(v_{spat}')$$

#### 4.2.2 3D RELATIONAL ATTENTION

Normal attention doesn't know about 3D positions. We fix this by adding position-based biases.

For each patch i, we make a 3D coordinate:

$$p_i = (x_i, y_i, d_i)$$

where $x_i, y_i$ are 2D positions (scaled to [-1,1]) and $d_i$ is the depth. For any two patches, we compute their distance in 3D:

$$\Delta p_{ij} = p_i - p_j$$

We split each dimension into 5 bins. So if $Delta x_{ij}$ it goes in bin 3 (the [0, 0.5] bin). We combine the three dimensions into one number:

$$ID_{ij} = bin_x + 5 \times bin_y + 25 \times bin_d$$

This gives us 125 possible relationships. Each gets its own learned bias that we add to attention:

$$\text{Attention}(Q, K, V) = \text{softmax}\left(\frac{QK^T}{\sqrt{d_k}} + B_{ij}\right)V$$

We stack 4 of these attention layers to get our final spatial features $F_{spat}$ .

### 4.2.3 COMBINING WITH SEMANTICS

The spatial features now understand 3D structure but don't know what they're looking at. We use cross-attention to let them query semantic information.

First, the semantic features go through the standard LVLM projector:

$$v_{sem}^{proj} = Projector(v_{sem})$$

Then spatial features attend to them:

$$F_{fused} = CrossAttention(Q = F_{spat}, K = v_{sem}^{proj}, V = v_{sem}^{proj})$$

We concatenate this with the original semantic features:

$$F_{concat} = [F_{fused} \| v_{sem}^{proj}]$$

And use a final MLP to merge everything:

$$F_{final} = MergerMLP(F_{concat})$$

These features go to the language model to generate text.

## 5 EXPERIMENTS

### 5.1 EXPERIMENTAL SETUP

**Architecture.**Our approach is built upon the LLaVA-1.5 7B model(Liu et al., 2024), which has demonstrated strong performance on multimodal understanding tasks. To enhance its 3D spatial reasoning ability, we integrate MiDaS v3.0(Birkl et al., 2023) as the depth estimation module. MiDaS is well known for robust zero-shot relative depth prediction across diverse scenarios. During training, only the projection layer and the LLM components within LLaVA-1.5 7B are updated.

**Training Dataset.**We train on the SpatialScore dataset(Wu et al., 2025), which contains 28,093 carefully designed spatial reasoning question–answer pairs. The dataset spans eight major categories: counting (10%), object localization (12%), 3D positional relations (15%), depth and distance (12%), object attributes (13%), camera and image transformations (13%), tracking (12%), and other spatial tasks (13%). This diverse task distribution allows the model to acquire broad spatial understanding.

**Evaluation Benchmark.**For rigorous assessment, we evaluate on the SpatialScore-Hard subset. This benchmark includes 1,400 challenging samples, selected by voting across 20 different MLLMs and further verified by human annotators. Although SpatialScore-Hard is derived from SpatialScore, we strictly ensure no overlap between training and test sets to prevent data leakage.

**Implementation Details.**Our model, DuoLLM, follows the public LLaVA-1.5 (7B)(Liu et al., 2024) framework. We adopt the two-stage training paradigm from LLaVA: vision–language feature alignment pretraining, followed by instruction fine-tuning.We adopt Low-Rank Adaptation (LoRA)(Hu et al., 2021)to efficiently fine-tune the large-scale pretrained model

Stage 1: Feature Alignment Pretraining. The goal is to align the outputs of our dual-stream vision backbone with the frozen LLM embedding space. We use a subset of the official LLaVA pretraining corpus. To reduce computation, we randomly sample 50k examples from the 550k dataset. Only the newly introduced projection layers, the 3D relational attention module, and cross-attention modules are trained in this stage.

Stage 2: End-to-End Instruction Fine-tuning. After alignment, we fine-tune on the SpatialScore dataset. This step strengthens the model's spatial reasoning abilities on top of the aligned representation. The vision encoder and depth estimator are kept frozen, while the remaining components, including parts of the LLM, are trained jointly in an end-to-end manner.

Table 1: Comparison of different models on various tasks.

| Methods | Overall | Count. | Obj.-Loc. | Pos.-Rel. | Dist. | Obj.-Prop. | Cam.&IT. | Tracking | Others |
|---|---|---|---|---|---|---|---|---|---|
| *7B and 8B models* | | | | | | | | | |
| LLaVA-OneVision-7B(Li et al., 2025) | 15.60 | 14.08 | 13.14 | 16.43 | 20.57 | 12.00 | 21.14 | 17.75 | 9.04 |
| Qwen2.5-VL-7B(Bai et al., 2025) | 15.21 | 4.93 | 5.71 | 20.56 | 17.71 | 8.00 | 21.14 | 21.30 | 19.43 |
| Cambrian-8B(Tong et al., 2024) | 15.43 | 10.57 | 21.14 | 21.93 | 14.43 | 18.86 | 20.57 | 11.42 | 22.23 |
| InternVL2.5-8B(Chen et al., 2025) | 13.00 | 7.04 | 21.14 | 11.62 | 12.57 | 10.86 | 20.57 | 8.86 | 12.23 |
| InternVL3-8B(Zhu et al., 2025) | 12.86 | 7.61 | 2.86 | 10.86 | 11.43 | 17.43 | 8.00 | 16.57 | 21.15 |
| **SpatialAgent Qwen-ReAct** | **30.29** | 10.56 | 41.71 | 63.55 | 34.86 | 23.43 | 46.29 | 16.57 | 13.14 |
| **SpatialAgent Qwen-PE** | **35.30** | 13.38 | 40.57 | 67.31 | 29.14 | 26.29 | 31.43 | 18.34 | 45.35 |
| **SpatialAgent Intern-ReAct** | **39.51** | 26.56 | 43.53 | 58.88 | 32.86 | 33.14 | 46.29 | 24.86 | 34.57 |
| *11B, 13B, and 14B models* | | | | | | | | | |
| LLaMA-3.2V-11B(Grattafiori et al., 2024) | 21.93 | 20.42 | 36.57 | 27.57 | 15.43 | 20.57 | 17.71 | 21.89 | 13.71 |
| LLaMA-3.2V-11B-CoT(Grattafiori et al., 2024) | 23.50 | 14.08 | 24.57 | 32.29 | 9.43 | 32.57 | 22.29 | 9.47 | 15.43 |
| LLaVA-1.5-13B(Liu et al., 2024) | 26.50 | 16.90 | 40.57 | 33.64 | 18.86 | 38.86 | 12.00 | 22.49 | 25.71 |
| SpaceLLaVA-13B(Chen et al., 2024) | 18.70 | 6.70 | 14.29 | 17.54 | 26.57 | 26.57 | 9.29 | 21.43 | 11.42 |
| InternVL3-14B(Zhu et al., 2025) | 16.14 | 14.08 | 7.43 | 12.62 | 20.00 | 20.00 | 19.43 | 10.06 | 25.71 |
| *32B and 38B models* | | | | | | | | | |
| Qwen2.5-VL-32B(Bai et al., 2025) | 14.36 | 11.27 | 11.43 | 14.49 | 20.00 | 6.29 | 14.29 | 25.44 | 11.43 |
| InternVL2.5-38B(Chen et al., 2025) | 11.64 | 7.04 | 10.29 | 13.55 | 21.71 | 6.86 | 7.43 | 15.38 | 9.71 |
| InternVL3-38B(Zhu et al., 2025) | 16.00 | 23.94 | 12.00 | 12.62 | 14.86 | 13.71 | 17.71 | 20.12 | 15.43 |
| *72B and 78B models* | | | | | | | | | |
| LLaVA-OneVision-72B(Li et al., 2025) | 15.29 | 16.20 | 8.00 | 12.62 | 18.86 | 18.29 | 21.71 | 15.98 | 11.43 |
| Qwen2.5VL-72B(Bai et al., 2025) | 17.79 | 9.86 | 22.86 | 20.56 | 19.43 | 26.29 | 21.71 | 21.31 | 18.86 |
| InternVL2.5-78B(Chen et al., 2025) | 13.43 | 11.27 | 19.43 | 11.21 | 15.43 | 4.00 | 17.71 | 14.20 | 14.29 |
| InternVL3-78B(Zhu et al., 2025) | 21.79 | 21.13 | 20.00 | 25.27 | 27.43 | 8.57 | 18.29 | 27.81 | 22.29 |
| *Proprietary Models (Commercial APIs)* | | | | | | | | | |
| Gemini-2.0 Flash | 28.92 | **28.87** | 40.57 | 23.36 | 21.71 | 21.71 | 23.43 | **49.70** | 24.00 |
| Claude-3.5-Haiku | 30.00 | 20.42 | 54.29 | 32.29 | 17.14 | 21.71 | 31.95 | 14.86 | 12.86 |
| GPT-4o | **30.57** | 23.94 | **54.86** | 29.91 | 24.57 | 25.71 | 24.57 | 39.05 | 21.14 |
| **DuoLLM(this paper)** | **34.57** | 10.56 | **70.29** | **48.13** | 20.00 | **38.86** | 26.29 | **36.09** | 19.43 |

## 5.2 MAIN RESULTS

**Overall Performance.** From Table1 we conduct a comprehensive evaluation of DuoLLM on SpatialScore-Hard. As shown in Table 1, our model achieves an overall accuracy of 34.64%, outperforming most open-source models of similar scale and even surpassing several strong proprietary systems. These results highlight the effectiveness of our approach.

**Comparison with Open-Source Models.** Among open-source models in the 7B/8B range, our method demonstrates competitive performance. While SpatialAgent-Intern-ReAct (39.51%) retains a slight advantage overall, our model shows clear strengths on specific tasks.Notably, in object localization, our model reaches 70.29% accuracy, dramatically higher than LLaVA-One-Vision-7B (13.14%) and Cambrain-8B (21.14%), and even surpassing the best-performing SpatialAgent variant (43.53%). This confirms the effectiveness of our dual-stream design and 3D perception module in addressing the fundamental "where" question in spatial reasoning. Similarly, in 3D positional relation reasoning, our model achieves 48.13%, comparable with state-of-the-art systems and well above baseline levels.

**Comparison with Proprietary Models.** Perhaps most strikingly, our model outperforms leading commercial APIs. As shown in the lower half of Table 1, DuoLLM (34.64%) exceeds GPT-4o (30.57%), Claude-3.5-Haiku (30.00%), and Gemini-2.0 Flash (28.92%). This suggests that, in specialized domains like spatial reasoning, a carefully designed open-source model with appropriate inductive biases can outperform much larger, general-purpose systems.

**Summary.** Across multiple challenging tasks, DuoLLM delivers results on par with or better than state-of-the-art baselines. Its particularly strong performance on object localization, together with its ability to surpass leading commercial models, demonstrates both the robustness and generalizability of our approach.

## 5.3 ABLATION STUDY

### 5.3.1 CORE ARCHITECTURE

Using LLaVA-1.5-13B as baseline, we first test the dual-stream backbone without the 3D engine (DuoLLM(ds)). This variant improves geometry-sensitive tasks such as positional relation (+9.82) and tracking (+2.95), confirming the benefit of separating spatial and semantic pathways. However, the overall gain is modest (+2.86) from Table2, indicating that dual-stream alone is insufficient.

Table 2: Ablation on Proposed Methods. ds and 3de refers to dual-stream and 3D engine information SpatialScore-Hard , respectively.

| Methods | Overall | Obj.-Loc. | Pos.-Rel. | Obj.-Prop. | Tracking |
|---|---|---|---|---|---|
| **LLaVA-1.5-13B** | 26.50 | 40.57 | 33.64 | 38.86 | 22.49 |
| **DuoLLM(ds)** | 29.34 | 50.29 | 40.65 | 30.29 | 25.44 |
| **DuoLLM(ds+3de)** | 34.57 | 70.29 | 48.13 | 38.86 | 36.09 |

Table 3: Ablation on 3D engine. dp , aa and ca refers to depth , 3D-relational-attention and cross-attention, respectively.

| Methods | Overall | Obj.-Loc. | Pos.-Rel. | Obj.-Prop. | Tracking |
|---|---|---|---|---|---|
| **3D engine(dp)** | 27.15 | 38.29 | 43.46 | 24.57 | 32.54 |
| **3D engine(dp+aa)** | 32.59 | 48.00 | 45.53 | 37.71 | 37.28 |
| **3D engine(dp+aa+ca)** | 34.57 | 70.29 | 48.13 | 38.86 | 36.09 |

Adding the full 3D perception engine (DuoLLM(ds+3de)) yields a major improvement: overall accuracy rises from 26.96 to 34.57 (+7.61), with object localization improving from 45.41 to 70.29 from Table2. This shows a clear synergy—the dual-stream backbone creates dedicated capacity for geometry, while the 3D engine provides essential priors.

### 5.3.2 COMPONENT ANALYSIS OF THE 3D PERCEPTION ENGINE

**Depth prior injection**. In this variant, monocular depth maps predicted by the depth estimator are injected into the spatial stream via element-wise addition. This baseline version already improves overall performance to 27.15, compared to 26.50 for LLaVA-1.5 without 3D information from Table3. This shows that compensating for the "missing dimension" with even simple geometric priors is a direct and effective way to boost performance.

**3D relational attention** . Adding relational attention on top of depth priors further raises overall accuracy to 32.59. This indicates that raw depth maps alone are insufficient; explicit modeling of relative spatial relations is critical. By incorporating geometric biases directly into attention, the model progresses from merely "seeing" depth to actually "reasoning" about 3D structure.

**Cross-attention fusion** . Finally, we introduce asymmetric cross-attention to fuse spatial and semantic streams. This yields the best performance, with overall accuracy reaching 34.57. The largest gain comes in object localization, which improves by over 22 points (from 48.00 to 70.29). These results validate our design choice of directional fusion: letting enhanced spatial features ("where") query semantic features ("what") ensures that spatial positions are consistently aligned with their semantic identities. This produces reliable joint representations for downstream language reasoning.

### 5.4 LIMITATIONS

DuoLLM's performance depends on its pretrained monocular depth estimator: while it leverages strong generalization without costly 3D annotations, its spatial reasoning may fail on images with transparent or highly reflective objects..

## 6 CONCLUSION

We introduce DuoLLM, a dual-stream multimodal architecture with an integrated 3D perception engine that breaks the cycle between semantic bias and missing geometric cues in LVLMs. On challenging benchmarks, it surpasses prior methods, with ablations showing that its strength lies in the synergy of semantic–spatial decoupling and 3D perception. This provides a principled path toward embedding geometric priors in LVLMs and advancing them toward true scene understanding for robotics and embodied AI.

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

# A APPENDIX

## CONTENTS

## MORE IMPLEMENTATION DETAILS

### HYPERPARAMETERS

The training of DuoLLM was carried out in two stages: feature alignment pre-training and instruction fine-tuning. All experiments were conducted on a single NVIDIA L20 GPU with bfloat16 mixed-precision training for about 14 hours.

**Stage 1: Pre-training.** In this stage, the goal was to establish an initial alignment between the vision encoder and the language model. Training was performed for one epoch using the AdamW optimizer with a learning rate of $1\times10$ and a weight decay of 0.05. The effective batch size was set to 32, achieved by using a per-device batch size of 8 with gradient accumulation over 4 steps. A cosine annealing learning rate schedule was adopted, with a warmup ratio of 3

**Stage 2: Fine-tuning.** For instruction tuning, we applied LoRA (Low-Rank Adaptation) to efficiently adapt the model to instruction-following tasks. Training was again conducted for one epoch with the AdamW optimizer. The learning rate was reduced to $1\times10$, while the weight decay remained at 0.05. The effective batch size was kept at 32. LoRA was configured with a rank of 16, an alpha of 32, and a dropout rate of 0.2. The cosine schedule was also used in this stage, with a smaller warmup ratio of 1

### DIFFERENT EPOCHS AS SHOWN IN TABLE4

Table 4: The accuracy rates on different epoch.

| Methods | Overall |
|---------|---------|
| **2epoch** | 20.06 |
| **1epoch** | 34.57 |
| | |

Dataset Size: The Hidden Bottleneck Accuracy plunges from 34.6% (1 epoch) to 20.1% (2 epochs), revealing a sharp small-data overfitting effect. With only 20K samples, DuoLLM quickly extracts useful spatial patterns in the first epoch but shifts to memorization in the second. The model's dual-stream design and 3D relational attention demand far larger datasets; under scarcity, its complexity accelerates overfitting rather than enabling stable convergence.

### LORA DROPOUT

LoRA Dropout: Balancing Regularization and Learning Figure 4 shows accuracy rising from 20.3% (dropout=0.05) to a peak of 31.4% (0.20), then slightly falling at 0.30. This indicates that moderate regularization is crucial: dropout=0.20 strikes the best balance between preventing overfitting and retaining learning capacity. The near-plateau beyond 0.20 suggests diminishing returns, with excessive dropout undercutting the model's ability to learn spatial relations. Given the small dataset ( 20K), careful dropout tuning is essential, with 0.20 emerging as the sweet spot for preserving DuoLLM's complex dual-stream reasoning while maintaining generalization.

### LORA RANK

Figure 5 highlights a sharp train–test divergence: training accuracy peaks at lora-r=16 (34.6%) but test accuracy is optimal at the much lower lora-r=0.2. Higher ranks improve memorization yet harm generalization, with accuracy even declining at lora-r=32 (31.4%). This reflects overfitting under the small-scale dataset ( 20K samples), where increased adaptation capacity captures training-specific patterns that fail to transfer. These results emphasize that DuoLLM requires low-rank calibration to preserve generalization and mitigate overfitting in dual-stream spatial reasoning.

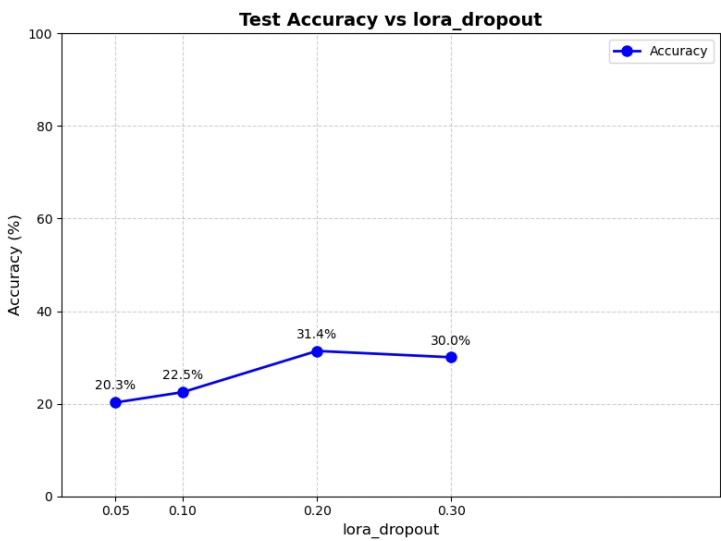

Figure 4: shows the accuracy rates among different lora-dropout.

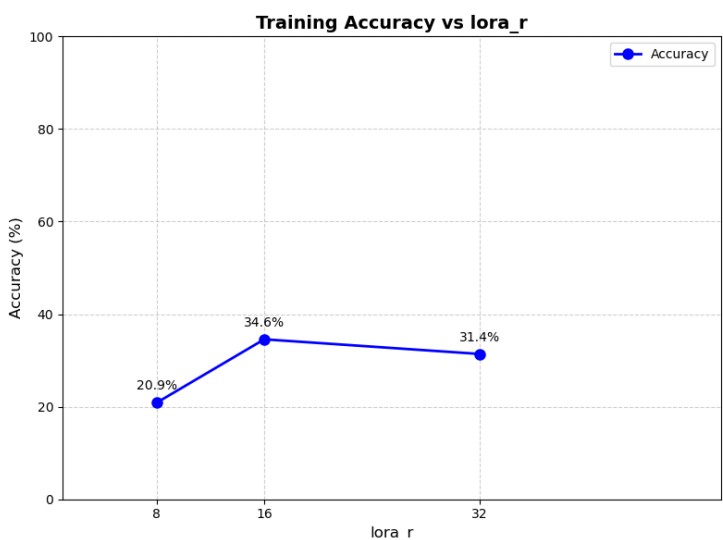

Figure 5: shows the accuracy rates among different lora-r.

DEPTH INTEGRATION: PROMISE AND PERILS

Table 5

| Methods | Overall | Obj.-Loc. | Pos.-Rel. | Obj.-Prop. | Tracking |
|---|---|---|---|---|---|
| **DuoLLM(ds)** | 29.34 | 50.29 | 40.65 | 30.29 | 25.44 |
| **3D engine(dp)** | 27.15 | 38.29 | 43.46 | 24.57 | 32.54 |

Interestingly, adding depth cues via the 3D engine does not consistently improve overall performance from Table 5 . While depth enhances spatial relation understanding (Pos.-Rel. +2.81), it degrades object localization and property recognition, likely due to noisy monocular depth estima-

tion and suboptimal fusion with semantic features. This suggests that naïve integration of depth signals may introduce noise rather than providing robust 3D priors.

THE USE OF LARGE LANGUAGE MODELSLLMS

We would like to acknowledge the invaluable assistance of state-of-the-art large language models, including ChatGPT, Claude, and Gemini. These tools have significantly supported our research process in various stages. In particular, they provided efficient information retrieval, offered timely debugging suggestions for code, and assisted in refining the clarity and readability of the manuscript. Their contributions have enhanced both the efficiency and quality of our work.

