# OpenReview forum: "DuoLLM: A Dual-Stream Decoupled Visual Language Model for 3D Spatial Reasoning"
_ICLR.cc/2026/Conference — ICLR 2026 Conference Withdrawn Submission_

### Official Review · Reviewer_hYK8 · 2025-10-26

**Soundness:** 2
**Presentation:** 1
**Contribution:** 1
**Rating:** 2
**Confidence:** 5

**Summary:**

This paper presents DuoLLM, a dual-stream large vision-language model (LVLM) explicitly designed for 3D spatial reasoning. Unlike single-stream LVLMs that intermix semantic and spatial information, DuoLLM introduces a decoupled architecture: a semantic stream for “what” and a spatial stream for “where.” The spatial stream integrates a 3D perception engine composed of (i) a monocular depth estimator, (ii) a 3D relational attention mechanism encoding geometric biases, and (iii) an asymmetric cross-attention module for semantic–spatial fusion.

**Strengths:**

1. The 3D perception engine effectively reintroduces depth and relative spatial priors into LVLMs, addressing a major limitation in prior single-stream models.
2. The dual-stream framework—separating semantic and spatial reasoning—is conceptually motivated and supported by neuroscientific analogies (“what/where” pathways).
3. Outperforms GPT-4o and Claude-3.5-Haiku in 3D reasoning, highlighting the value of domain-specific inductive bias even at smaller model scales.

**Weaknesses:**

1. The paper’s presentation is too poor, with many grammatical mistakes and severe formatting issues between text and figures. The current version does not meet ICLR standards. The authors should thoroughly revise the writing and layout for clarity and professionalism.
2. The idea of decoupling semantic and geometric features has been well explored in VG-LLM (NeurIPS 2025), Inst3D-LLM (CVPR 2025), and Spatial-MLLM (arXiv 2025). These works should be included and compared to better position this paper.
3. Experiments are only conducted on SpatialScore-Hard. Please include more benchmarks (e.g., CLEVR, GQA, ScanRefer, Multi3DRefer) to demonstrate generalization and robustness.

**Questions:**

See the weaknesses.

---

### Official Review · Reviewer_nXfk · 2025-10-31

**Soundness:** 2
**Presentation:** 1
**Contribution:** 2
**Rating:** 2
**Confidence:** 3

**Summary:**

This paper introduces DuoLLM, a dual-stream Vision-Language Model (VLM) designed to improve 3D spatial reasoning. The authors identify two core problems in existing VLMs: "Semantic Prioritization," where semantic understanding overshadows geometric analysis, and "3D Information Loss" from projecting 3D scenes onto 2D images. DuoLLM's architecture explicitly decouples processing into a semantic stream and a spatial stream. The spatial stream is enhanced with a "3D perception engine" that integrates depth information from a pre-trained monocular depth estimator, employs a novel 3D relational attention mechanism to encode relative positions, and uses cross-attention to fuse spatial features with semantic information. The model is trained and evaluated on the SpatialScore dataset and its "Hard" subset, where it shows improved performance over baseline VLMs and some proprietary APIs, particularly in object localization tasks.

**Strengths:**

1. **Clear Problem Formulation:** The paper does a good job of identifying and explaining two well-known and critical challenges in VLM spatial reasoning: the tendency for semantic features to dominate ("Semantic Prioritization") and the loss of geometric cues in 2D projections ("3D Information Loss"). The motivation for the work is clear.
2. **Logical Architectural Design:** The core architectural idea of decoupling semantic and spatial processing streams is intuitive and well-motivated by concepts from neuroscience (the "what" and "where" pathways). Creating a dedicated pathway for geometric information is a logical approach to address the identified problems.
3. **Thorough Ablation Study:** The ablation studies presented in Tables 2 and 3 are a highlight of the paper. They systematically validate the contribution of each component of the proposed architecture, demonstrating that the dual-stream design, depth priors, 3D relational attention, and cross-attention fusion all contribute positively to the final performance.

**Weaknesses:**

1. **Critically Flawed Evaluation:** The paper's most significant weakness is its evaluation protocol. The model is trained on the SpatialScore dataset and evaluated on SpatialScore-Hard, which is a curated, challenging subset of the same dataset. While the authors state there is no direct sample overlap, evaluating on the same data distribution from which the model was trained provides very weak evidence of generalization. The claims of the model's effectiveness are not credible without rigorous testing on multiple, diverse, out-of-domain benchmarks for spatial reasoning (e.g., VSI-Bench, ViewSpatial-Bench, SPAR-Bench).
2. **Poor Presentation and Lack of Clarity:** The paper suffers from numerous presentation issues.
    - **Figures:** The figures (especially 1, 2, and 3) are low-quality, simplistic, and lack the professionalism expected for a top-tier conference. Figure 3, the overall framework diagram, is cluttered and difficult to interpret.
    - **Writing:** The writing is often informal, imprecise ("features get pushed aside," "it seemed to work best in our experiments"), and contains grammatical errors.
    - **Missing Details:** The methodology section lacks critical details and justifications for key design choices. For example, the selection of layer -16 for spatial features is not justified, and the binning strategy for the 3D relational attention is poorly explained and not validated.
3. **Overstated and Unclear Novelty:** The paper claims that its 3D relational attention mechanism is the "first attempt in LVLMs to deeply integrate externally predicted geometric priors into the reasoning core in a structured way." This is a strong and likely inaccurate claim, as many recent works have explored integrating geometric information into attention mechanisms. The core components—multi-layer feature extraction, using off-the-shelf depth estimators, and relative positional biases—are extensions of existing and well-known techniques, not fundamental breakthroughs.
4. **Incomplete Performance Analysis:** On its chosen benchmark, DuoLLM (34.57%) is significantly outperformed by the specialized `SpatialAgent` models (e.g., 39.51%). The paper fails to provide any meaningful discussion or analysis for why its supposedly superior architecture does not surpass the state-of-the-art. Furthermore, the performance is highly skewed, with exceptional results in object localization but very poor performance in other categories like counting and distance estimation, a limitation that is not discussed.

**Questions:**

1. Why was the evaluation restricted to the SpatialScore-Hard benchmark? Given that robust generalization is a key goal, why were out-of-domain benchmarks not used to validate the model's capabilities?
2. Could you provide a clearer explanation and justification for the design of the 3D relational attention? Specifically, how were the 5 bins for each spatial dimension defined, and was any analysis done to determine if this level of quantization is optimal?
3. The paper claims to be the "first" to integrate geometric priors into the attention core in this way. Could you please clarify this claim by comparing your approach to other works that have also incorporated 3D geometry or depth information into VLM attention mechanisms?
4. Your model is substantially outperformed by the `SpatialAgent` models on the same benchmark. What architectural or methodological limitations in DuoLLM do you believe are responsible for this performance gap, despite your model's theoretically stronger inductive biases for 3D?

---

### Official Review · Reviewer_7CVz · 2025-10-31

**Soundness:** 2
**Presentation:** 2
**Contribution:** 2
**Rating:** 2
**Confidence:** 3

**Summary:**

This paper investigates the limitations of current vision-language models in performing 3D spatial reasoning. The authors identify two key causes of failure: semantic bias, where high-level semantic features dominate over geometric understanding in single-stream architectures, and 3D information loss, which arises from projecting inherently 3D scenes into 2D inputs. To address these issues, the paper introduces DuoLLM, a dual-stream multimodal model that explicitly separates semantic and spatial processing. The model incorporates a 2.5D perception engine based on monocular depth estimation to recover geometric priors, a 3D relational attention mechanism to model object-to-object spatial relationships, and an asymmetric cross-attention module that fuses spatial and semantic features. Built on top of the LLaVA-1.5-7B framework and trained on the SpatialScore dataset, DuoLLM is evaluated on the SpatialScore-Hard benchmark. It achieves 34.6% accuracy, surpassing most open-source baselines and outperforming proprietary models such as GPT-4o and Gemini-2.0 Flash on spatial reasoning tasks. Ablation studies demonstrate that both architectural decoupling and 3D perception are critical for achieving robust spatial reasoning, supporting the authors’ claim that explicitly modeling geometric priors is a promising path toward extending LVLMs beyond 2D recognition.

**Strengths:**

1. The paper presents a well-defined motivation by identifying the “vicious cycle” underlying spatial reasoning failures in large vision-language models (LVLMs). It clearly articulates how architectural biases and the inherent information loss during 3D-to-2D projection jointly degrade geometric understanding, offering a coherent conceptual framing for the proposed intervention.

2. The proposed dual-stream design is straightforward and easy to follow, making the overall contribution accessible while remaining technically meaningful. Compared to prior works, the model introduces a specialized 3D relational attention mechanism for injecting explicit geometric awareness into the representation learning process. This addition provides a concrete means of embedding structured 3D inductive biases within the VLM pipeline.

**Weaknesses:**

1. While the motivation is clear, the core idea—injecting structural geometric priors or depth cues into VLMs through inductive biases—is not entirely novel. Similar approaches have been explored in recent studies such as SpatialRGPT (NeurIPS 2024), SSR: Enhancing Depth Perception in Vision-Language Models via Rationale-Guided Spatial Reasoning (2025), and LLaVA-3D (2024), all of which integrate depth or 3D structure into LVLMs. The present work extends these ideas but does not sufficiently differentiate itself in methodological innovation or analysis.

2. The evaluation does not include comparisons with some of the strongest recent baselines, particularly the Gemini 2.5 series and other leading multimodal foundation models. Moreover, the overall performance remains below that of top-performing open-source spatial reasoning frameworks such as SpatialAgent. Additional experiments examining how depth quality (e.g., from different depth estimators) affects performance would provide stronger evidence for the robustness of the proposed 3D perception module.

3. The figures lack clarity and sufficient captioning to guide interpretation. Some visualizations do not effectively convey the key insights of the architecture or results, and the tables contain ambiguous or inconsistent notation that can confuse readers.

4. The related work section is somewhat simple and does not sufficiently cover expanding literature on 3D-aware VLMs and spatial reasoning models published in 2024–2025. In particular, several relevant contemporaneous works addressing depth integration, 3D reconstruction, and relational grounding are missing. A deeper comparative discussion would better situate this paper within the evolving research landscape. A few examples:
[1] Spatial-MLLM: Boosting MLLM Capabilities in Visual-based Spatial Intelligence
[2] SpatialReasoner: Towards Explicit and Generalizable 3D Spatial Reasoning
[3] MM-Spatial: Exploring 3D Spatial Understanding in Multimodal LLMs
etc.

**Questions:**

Please refer to weaknesses.

---

### Official Review · Reviewer_ndcM · 2025-11-01

**Soundness:** 1
**Presentation:** 1
**Contribution:** 1
**Rating:** 2
**Confidence:** 5

**Summary:**

The paper studies how to better incorporate depth and spatial cues into VLMs to improve 3D spatial reasoning. Specifically, the paper introduces a relational attention module to capture object-to-object spatial relationships and an asymmetric cross-attention module to fuse spatial features with semantic representations.

**Strengths:**

The paper addresses an important open challenge in making VLMs more vision-centric, specifically how to design the vision encoder so that spatial structure is preserved rather than overwhelmed by semantic features.

**Weaknesses:**

* The paper is difficult to follow. The writing introduces many unnecessary new terms, and the presentation quality is low (e.g., flawed formatting, blurred figures without clear purpose). Overall, it feels far from a publish-ready state.
* It is unclear how the proposed relational attention reliably captures object-to-object spatial relations, as no explicit instance-level cues (e.g., segmentation or proposals) are provided. The method seems to rely on emergent separation in feature space, which is not guaranteed.
* The paper lacks comparisons to simple feature fusion baselines (e.g., concat, element-wise addition, standard cross-attention, perceivers, etc). Without these, it is unclear whether the proposed method offers improvements beyond basic fusion methods.
* The experimental results are limited to a single, not widely adopted benchmark, making it difficult to judge generality or robustness.

**Questions:**

Please see the weakness section.

---

### Note · Authors · 2025-11-12

I have read and agree with the venue's withdrawal policy on behalf of myself and my co-authors.